# Shape Memory Polyurethane Biocomposites Based on Toughened Polycaprolactone Promoted by Nano-Chitosan

**DOI:** 10.3390/nano9020225

**Published:** 2019-02-07

**Authors:** Arvind Gupta, Beom Soo Kim

**Affiliations:** Department of Chemical Engineering, Chungbuk National University, Cheongju, Chungbuk 28644, Korea; myarvind2003@gmail.com

**Keywords:** chitosan, polycaprolactone, shape memory, stretchability, polyurethane, biocomposite

## Abstract

The distinctive ability to remember their original form after partial or complete deformation makes shape memory polymers remarkable materials for several engineering and biomedical applications. In the present work, the development of a polycaprolactone based toughened shape memory polyurethane biocomposite promoted by in situ incorporation of chitosan flakes has been demonstrated. The chitosan flakes were homogeneously present in the polymer matrix in the form of nanoflakes, as confirmed by the electron microscopic analysis and probably developed a crosslinked node that promoted toughness (*a* > 500% elongation at break) and led to *a* ~130% increment in ultimate tensile strength, as analyzed using a universal testing machine. During a tensile pull, X-ray analysis revealed the development of crystallites, which resulted from a stress induced crystallization process that may retain the shape and melting of the crystallites stimulating shape recovery (with *a* ~100% shape recovery ratio), even after permanent deformation. The biodegradable polyurethane biocomposite also demonstrates relatively high thermal stability (*T_max_* at ~360 °C). The prepared material possesses a unique shape memory behavior, even after permanent deformation up to *a* > 500% strain, which may have great potential in several biomedical applications.

## 1. Introduction

The distinctive ability to remember their original form after partial or complete deformation makes the shape memory polymers (SMP) remarkable materials for several engineering and biomedical applications. Along with the widely known shape memory alloy, Nitinol (Nickel-Titanium Naval Ordnance Laboratory) [1], various polymers such as polyacrylates [2], polyimides [3], styrene-butadiene copolymers [4,5], polyurethane (PU) [6], polystyrene [7], etc., are known to have a shape memory ability on application of an external stimulus, such as pH, electrical, mechanical, hydration, magnetic, heat, light, etc. [8,9]. Among these polymers, polyurethane is found to be a versatile material due to its extraordinary characteristics, such as its relatively high toughness, light weight, tolerance to strain, high strength and elongation at break, abrasion resistance, low cost, and easy processability; polyurethane has applications in various domains, ranging from electronic packaging, food packaging, biomedical, aerospace, automobiles, textiles, energy, etc., to commodity applications [10,11].

Polyurethanes are a class of polymers with inherent two-phase separated domains due to a microphase inhomogeneity containing a urethane bond between an alternating polymer block of hard and soft segments, wherein diisocyanate is a hard segment connected to the flexible soft segments made of polyether or polyester diol chains. The presence of a hard segment is responsible for remembering the original shape, whereas the soft segment stores the energy for dissipation and provides the desired force, allowing polyurethane to return to its original form. Herein, the soft segments, i.e., polyol, can broadly be classified as carbonate based (i.e., polyhexamethylene carbonate diol) [12], ether based (i.e., polytetrahydrofuran diol) [13], and ester based (i.e., polycaprolactone diol) [14]. These conventional polyols are generally obtained from petroleum-based resources, and only some of them are biodegradable [15].

It is now recognized by the scientific community that the biopolymers, biobased and biodegradable polymers, have the potential to replace existing petroleum based non-biodegradable polymers and may thus be considered promising future materials [16]. These polymers include polylactic acid (PLA), polycaprolactone (PCL), polyglycolic acid (PGA), polyhydroxybutyrate (PHB), etc., which have been found to be useful in applications ranging from biomedical to automobile, along with day to day applications according to the properties of the polymers [17]. For several areas, such as tissue engineering, orthopedic implants, or other biomedical applications, PCL has been found to be one of the most promising polymeric materials due to its lower melting temperature, easy processability, biodegradability, lower glass transition temperature, nontoxicity, relatively higher thermal degradation temperature, etc. [18]. However, it has been found that PCL has inherently unique characteristics, and certain limitations of this polymer restrict its direct use in the targeted applications [19]. The desired applications often require specific properties which are difficult to determine from a pristine polymer.

In order to serve their purpose, biodegradable polymers such as PCL can be used in combination with each other. Moreover, different techniques, such as blending of polymers [20], copolymerization [21], fabrication of biocomposite with biofillers [22,23], grafting [24], crosslinking [25], etc., can be used to tailor the properties of the PCL based final product. Fabrication of the composite with fillers via blending, in situ polymerization [26], and reinforcement [27], is a promising approach to tailor the final product’s properties, such as mechanical strength, toughness, stiffness, biodegradability, and behavior of shape memory. Several organic and inorganic materials, such as carbon based fillers, chitosan [28], lignin [29], graphene [30], cellulose [31], magnetic nanoparticles [32], and metal nanoparticles [33] can be used as fillers for the development of biodegradable polyurethanes. Among these fillers, chitosan, which is often used as both filler and polymer matrix, is a promising biopolymer. Chitosan, which can be utilized in biomedical applications, is recognized as nontoxic, biodegradable, and antibacterial in nature. Chitosan can be produced using a process of chitin deacetylation, which is derived from the bioresource.

It is known that chitosan has a melting point near degradation temperature, which restricts its thermal processing both in its pristine form and with other polymers. The presence of free amino functional groups in the chitosan molecule aids in solubilizing it in acidic media by functional protonation. Additionally, it is challenging to obtain a uniform distribution of chitosan throughout the polymer matrix due to its hydrophilic nature. Several researchers have developed different process to modify chitosan and use it in biocomposite fabrication [34,35]. Various research groups are working worldwide to utilize chitosan in shape memory applications. By incorporating bioactive glass nanoparticles obtained from the sol-gel process, Leite et al. [36] have synthesized chitosan spheres. They reported a shape memory effect triggered by hydration and used for healing bones and controlling drug release. Bao et al. [37] have developed chitosan functionalized poly(lactic-co-glycolic acid) (PLGA) microspheres containing lysozyme. They have used high intensity focused ultrasound to stimulate the shape memory effect. Zhang et al. [38] used microwaves as a shape memory stimulant in PCL foam and found improvement in the recovery speed. Similarly, Shou et al. [39] have developed near infrared light (NIR) responsive shape memory films via photo-initiated polymerization of PCL with acryloyl terminal groups in the presence of gold nanorods. They conceptualized a controllable shape memory PCL, embedded with a gold nanorod film, that can actuate in a physiological temperature using NIR situated at a remote place. However, as per our knowledge, the use of chitosan as a filler in a PCL matrix by in situ reaction for a shape memory application has not been explored.

Therefore, the present work is focused on the development of a thermo-responsive shape memory biodegradable PCL based polyurethane and its composites. Herein, the chitosan flakes (nm to µm scale) react with the PCL diol, thereby providing a crosslinking point in the polymer matrix, which potentially enhances the shape memory effect along with the mechanical properties of the PCL based polyurethane. Several techniques, such as universal testing machine (UTM), thermogravimetric analysis (TGA), differential scanning calorimetry (DSC), fourier transform infrared (FTIR), and X-ray diffraction (XRD) have been used to characterize the prepared polyurethane biocomposites.

## 2. Materials and Methods

### 2.1. Materials

Chitosan was derived from shrimp shells with ≥75% deacetylation, which were obtained from Sigma Aldrich (Reykjavik, Iceland). PCL diol with an average molecular weight (Mn) of 2000 g/mole was purchased from Sigma Aldrich (Hong kong, China). 1,6-hexamethylene diisocynate (HMDI) was supplied by Daejung (Siheung, Korea). Dibutyltin dilaurate (DBDTL), used as catalyst, was provided by Sigma (St. Louis, MO, USA). Acetone and chloroform were procured from J.T. Baker Avantor (Center Valley, PA, USA). N,N-Dimethylformamide (DMF) and 1,4-butanediol (BDO) were purchased from Tedia Company Inc. (Fairfield, OH, USA) and Aldrich (Steinheim, Germany), respectively. Hydrogen peroxide (H_2_O_2_) was received from Samchun (Seoul, Korea). All the received chemicals were used without further purification, unless otherwise stated.

### 2.2. Preparation of Polyurethane Biocomposite

It is known that chitosan is available in the form of relatively large flakes and insoluble in DMF, due to strong glycosidic linkage in the backbone molecules. Initially, 10 g of chitosan was added to 100 mL of hydrogen peroxide (30%) by stirring (MS-DMS633, Mtops, Yangju, Korea) at 40 °C for 24 h to increase the dispersion and allow for swelling. After swelling, the prepared slurry was added to a sufficient amount of acetone to precipitate the chitosan. The swelling and precipitation processes are known to reduce the size of the obtained flakes and weaken glycosidic linkages. The obtained purified chitosan was washed with an adequate amount of deionized water several times until a neutral pH was obtained; this process was followed by freeze drying (IIshin Lab Co. Ltd., Seoul, Korea) for 24 h. The resulting chitosan (the required amount) was allowed to swell in 10 mL of DMF for 24 h in order to use it for further biocomposite fabrication.

The Chitosan promoted PCL based shape memory polyurethane biocomposite was prepared according to the following steps. The required amount of PCL diol was dropped into a two-neck round-bottom flask equipped with a magnetic stirring bar and an inert environment kept at 40 °C on a stirrer with a hot mantle. Subsequently, the predetermined amount of chitosan swollen in DMF was added to the flask to maintain at 5%, 7.5%, 10%, and 20% against the PCL. After dissolution of the PCL, the temperature was raised to 80 °C, and stirring was maintained at 200 rpm. HMDI was added to the solution dropwise under inert conditions, along with a few drops of DBTDL as a catalyst. The reaction continued for 2 h to form a diisocynate terminated prepolymer. BDO was added to the reaction system as a chain extender, and stirring continued. The chain extension was followed by the addition of chloroform under a reflux condition to reduce the viscosity of the reaction system. The solution was then poured on a polypropylene tray. The solution tray was kept in a fume hood to allow the evaporation of the chloroform, followed by drying in a vacuum oven at 80 °C for 24 h. Additionally, PCL polyurethane without chitosan was prepared by following the same procedure as described above for the comparison. All samples was prepared by maintaining the NCO:OH final ratio as 1:1.1. The obtained film was stored in a refrigerator for further characterization. The polyurethane biocomposites with 5%, 7.5%, 10%, and 20% chitosan content were denoted as PCL-PU-M5, PCL-PU-M7.5, PCL-PU-M10, and PCL-PU-M20, respectively, while neat PCL polyurethane was denoted as PCL-PU.

### 2.3. Characterization

Fourier transform infrared (FTIR) spectroscopy spectra were obtained on a Nicolet IR 200 (Thermo Scientific, Waltham, MA, USA) instrument at room temperature. The spectra of the thin film were recorded against air as reference in the range between 4000 and 600 cm^−1^ after 64 scans with 4 cm^−1^ resolution.

Differential scanning calorimetry (DSC Q2000, TA Instruments, New Castle, DE, USA) was used to measure melting, crystallization, and glass transitions of the sample under an inert environment. The pre-weighed sample (5–10 mg) was heated from 25 °C to 150 °C with a rate of 10 °C/min and kept under isothermal conditions for 5 min to eliminate its thermal history. The sample was then cooled to 25 °C with a rate of 10 °C/min, kept isothermal for 5 min, and heated again to 150 °C with the same rate.

Thermogravimetric analysis (D-TGA, SDT 2960, TA Instruments) was carried out by heating 2–5 mg of the sample from room temperature to 500 °C with a rate of 10 °C/min under a N_2_ environment.

X-ray diffraction (XRD) spectra were recorded using an automated multipurpose X-ray diffractometer (JP/SmartLab, 9kW, Rigaku, Tokyo, Japan) equipped with copper (Cu) Kα radiation (1.540593 Å) as an X-ray source (40 kV, 200 mA) with a monochromatic filter along with a detector (SC-70). The samples were analyzed in the range of 5–30° at a scan rate of 2°/min with a 0.02° resolution.

Contact angle measurement equipment (Model GSM) supplied by Surfacetech Co. Ltd. (Gwangju, Korea) was used to measure the contact angle of 2 µL of deionized water dropped on the surface of the prepared biocomposites. The films were pasted on a cleaned glass slide prior to measurement, and the angle was determined after 2 min of drop stabilization. The contact angles were determined from at least three different regions on each sample surface along with standard deviation.

Tensile strength and percent elongation of the prepared biocomposite samples were analyzed using a universal testing machine (UTM) (LRK-5kN, Cometech, QC-M2014, NTS Technology Co. Ltd. Chengdu, China). The specimens (5 mm width, 0.5–1.0 mm thickness, and 20 mm gauge length) were fixed and analyzed under a UTM equipped with a 5 kN load cell at a fixed cross-head speed of 5 mm/min in tensile mode. The data obtained were analyzed using AMIS 1.1.6 software and reported as the mean of at least five replicates of each sample along with standard deviation.

The topography of the fractured samples was examined using a field emission scanning electron microscope (FESEM). The samples were gold coated in a sputtering unit and characterized using FESEM (Ultra Plus, Zeiss GmbH) at an accelerating voltage of 3 kV after placing it on a carbon tape. The micrograph was analyzed using Gwyddion version 2.25 software (Cszech Metrology Institute, Brno, Cszech Republic).

The gel content, formed due to the formation of a cross-linked three-dimensional network, was quantified using Equation (1):(1)$$\mathrm{Gel}\;\mathrm{content}\left(\%\right)=\frac{W_{gel}}{W_{i}}\times 100$$
where *W_i_* and *W_gel_* represent the initial weight of the prepared biocomposite and the weight of dried gel, respectively, after washing with chloroform and then vacuum drying.

The shape memory behavior of the prepared biocomposite was analyzed by performing a tensile test in warm water (50–60 °C) and hot oven (50 °C) conditions. The test was conducted in two environments: (1) Tensile pull and recovery in warm water, and (2) shape setting and recovery in a hot air condition. The sample was deformed by a tensile pull at 50 °C until a 100% strain was achieved. Next, the stretched shape was fixed by cooling the sample to 20 °C for 5 min, and the length was measured. The stretched and fixed sample was dropped in warm water (50–60 °C), and the length was measured after the recovery of the sample from the water. The shape fixing ratio and shape recovery ratio were calculated as per the following equations:(2)$$\mathrm{Shape}\;\mathrm{fixing}\;\mathrm{ratio}\left(R_{f}\right)=\frac{\left(L_{s}-L_{i}\right)}{L_{i}}\times 100$$
(3)$$\mathrm{Shape}\;\mathrm{recovery}\;\mathrm{ratio}\;\left(R_{r}\right)=\frac{\left(L_{o}-L_{f}\right)}{L_{i}}\times 100$$
where *L_o_*, *L_s_*, *L_f_*, and *L_i_* represent the length of the sample after stress, the length of the sample after the release of stress, the length of the recovered sample, and the initial length of the sample, respectively.

## 3. Results and Discussion

### 3.1. Synthesis of the PCL Based Polyurethane Biocomposite

The molecular functional groups in the PCL-PU and PCL-PU-chitosan biocomposites were examined using FTIR. The spectra, shown in Figure 1, display the characteristic peaks for the PCL-PU and chitosan. The PCL-PU characteristic peaks at 1732 cm^−1^, 2947 cm^−1^, and 3332 cm^−1^ are assigned to the stretching vibration of the carbonyl group, the aliphatic C–H bond, and the N–H bond, respectively [40]. The band at 3442 cm^−1^ corresponds to the N–H and OH stretching vibration, whereas peaks at 2947 cm^−1^ and 2868 cm^−1^ are related, respectively, to the C–H symmetric and asymmetric stretching in chitosan. The bands around 1658 cm^−1^, 1571 cm^−1^, and 1326 cm^−1^ correspond to the C=O stretching of amide I, the N–H bending of amide II, and the C–N stretching of amide III of the chitosan acetyl group, respectively. The peaks merged between 1195 cm^−1^ and 936 cm^−1^ are related to the asymmetric stretching of the C–O–C backbone and the skeletal C–O stretching in pristine chitosan, respectively [41]. In the case of PCL-PU and other biocomposites, the peaks corresponding to the C–N stretching and N–H in-plane bending in the C–N–H functional group are located between 1643 cm^−1^ and 1504 cm^−1^. A shoulder found on the absorption peak is attributed to the carbonyl group in PCL-PU. The peak at 1732–1724 cm^−1^ corresponds to the vibration of the amide group due to the urethane block and chitosan molecules. The stretching band at 3520 cm^−1^ corresponds to the hydroxyl functional group in PCL diol (not shown in the figure), which disappeared, while a new band, attributed to the N–H stretching, appears at 3332 cm^−1^, thereby confirming the formation of polyurethane. The presence of peaks related to PCL and chitosan in the prepared biocomposites confirms the formation of polyurethane and chitosan biocomposites.

### 3.2. Thermal Transitions of Prepared Polyurethane Biocomposites

The prepared PCL based polyurethane biocomposites were analyzed using DSC, and the thermograms are shown in Figure 2. The heating cycle (Figure 2a) of the biocomposite shows the presence of PCL in crystalline form, with a melting temperature around 40 °C. The melting temperature of biocomposites is found to be the same with PCL-PU, whereas the enthalpy of melting is reduced from 23.7 J/g to 11.0 J/g as the content of chitosan in the polymer matrix is increased. The reduction in the melting enthalpy suggests a reduction in the crystallinity of the PCL with increasing chitosan content. The presence of chitosan possibly hinders PCL chain mobility, thereby restricting the orientation of chains and forming a crystalline domain, which may be responsible for the reduction in the crystallinity of the biocomposite. Chitosan nano-sized flakes are connected to PCL chains, which form the crosslinking point in the polyurethane system, resulting in an increased amorphous domain. In the case of 20% chitosan content, the melting temperature of the biocomposite is higher (46.7 °C), which could be attributed to the phase separation of the PCL and chitosan molecules, and the crystalline domain is reduced to 11.0 J/g. The exothermic peak of the biocomposite in the cooling cycle is found to be shifted towards a lower temperature, as presented in Figure 2b. The reduction in the melt crystallization temperature from 2.0 °C for PCL-PU to −9.6 °C for PCL-PU-M10 suggests that the PCL chains in biocomposites need less energy to be crystallized. However the enthalpy of crystallization is reduced from 22.6 J/g to 8.3 J/g, which could be due to the anti-nucleation effect of the chitosan flakes.

### 3.3. X-ray and Contact Angle Measurements

The biocomposite was characterized using X-ray diffraction spectroscopy, as shown in the Figure 3a. The characteristic peaks for PCL at 21.3°and 23.6° correspond to the (110) and (200) planes of the PCL crystal, whereas the peak at 19.3° is related to the (110) plane of the chitosan micro crystalline domain reflection [42,43]. The peak corresponding to chitosan has a significantly lower intensity in PCL-PU polyurethane biocomposites, whereas it is visible in the biocomposite with 20% chitosan content, indicating the presence of phase separated domains. No peak shifting was observed, indicating no change in the crystal structure of the PCL and chitosan.

Contact angle measurement of the prepared biocomposites was employed to understand their interaction with water molecules. The data are presented in Figure 4, and the contact angle was found to increase from around 69° to ~95°, alongside the increase in the content of chitosan from 0% to 10%. It is known that wettability is based on the chemical composition, surface topography, and surface free energy of the substance. As the chitosan flake content increases in the biocomposite, the contact angle gradually increases, possibly due to the reduction in surface free energy. The presence of chitosan flakes also affects the surface roughness, which may lead to an increase in the water contact angle. It is observed that the contact angle reduces to ~80° in a case of 20% chitosan, suggesting phase separation and a reduction in hydrophobicity.

### 3.4. Mechanical Properties of Polyurethane Biocomposites

The presence of nano-sized chitosan flakes influences the mechanical properties of the prepared polyurethane biocomposites. It is found that chitosan nanoflakes provide crosslink sites, thereby providing strength and imparting shape memory properties to the polyurethane biocomposite. The representative load-elongation curve and ultimate tensile strength (UTS), along with the elongation at break for PCL-PU and PCL-PU-Chitosan polyurethane biocomposites, are shown in Figure 5a,b. The UTS for PCL-PU was found to be 5.2 MPa, which increases by approximately 130% and reaches 12 MPa for the PCL-PU-M10 composite. The strain hardening due to the strong interfacial bonding between the chitosan and PCL chains and the presence of chitosan nano-sized flakes possibly resist the mechanical pull after a certain elongation, ultimately resulting in increased UTS. The elongation at break for PCL-PU was found to be 102%, which increases to 598% after the addition of 10% chitosan in the polyurethane biocomposite, indicating a significant improvement in the ductility of the PCL-PU biocomposite with integration of chitosan flakes. The increase in elongation could be the result of the uncoiling of coiled PCL chains present in the polyurethane biocomposite. It is also known that enhancement in the elongation at break or ductility delays the fracture, which reduces the risk of abrupt mechanical failure of the polyurethane biocomposite. As reported [44], the use of chitosan as a filler may improve the UTS of the polymer matrix with compromising elongation at break, whereas an increase in tensile strength along with elongation at break is observed in the present study. Increased elongation at break could be due to the chemical interaction of chitosan nanoflakes with PCL chains, which increase the coiling of the PCL chains at particular crosslink sites. The integration of chitosan with PCL chains can act as a bridge between the uncoiling and sliding chains, thereby lengthening the breakage process. However, in the case of PCL-PU-M20, the elongation at break is reduced to 5.8%. The increased content of chitosan (20%) in the PCL based polyurethane matrix ensures an improvement of ~21% in tensile strength compared to PCL-PU. However, the presence of chitosan in higher amounts may induce voids during the elongation process, ultimately resulting in breakage and a rupture of bonds leading to a reduction in the elongation at break.

### 3.5. Morphological Analysis of Polyurethane Biocomposites

The dispersion of the chitosan flakes in the PCL polyurethane matrix resulted in improved mechanical properties, which were analyzed using FESEM. A micrograph of the fractured surface of the PCL-PU and PCL-PU-Chitosan biocomposites, which displays a homogeneous dispersion of the chitosan flakes, is shown in Figure 6. It can be seen from the figure that the chitosan flakes reduced in size during biocomposite fabrication and were found to be around ~150 nm in width and ~600–1000 nm in length, whereas the PCL-PU has a relatively smooth surface.

### 3.6. Thermal Degradation of Polyurethane Biocomposites

The thermal behavior of the prepared polyurethane biocomposites was analyzed using TGA and presented in Figure 7. The onset degradation temperature of the chitosan was found to be 135 °C, whereas the maximum degradation rate was observed around 235 °C. It can be surmised that the presence of chitosan may affect the degradation behavior of the biocomposite due to its relatively lower degradation temperature. Conversely, no adverse effect was found in the biocomposite due to the presence of chitosan. The data for the degradation temperature and residue content is listed in Table 1. The onset degradation temperature of the polyurethane biocomposite increased from 310 ºC to 346 °C, alongside the increase in the content of chitosan from 5% to 10%. Similarly, the maximum degradation temperature improved from 360 °C to 384 °C, with an upsurge in the content of chitosan from 0% to 10%. An increase in the content of chitosan to 20% lead to a reduction in onset degradation (327 °C) as well as maximum degradation temperature (379 °C). The effect of chitosan in the present polyurethane biocomposite can be better understood by estimating the degradation temperature at 10% weight loss (T_10_). In Table 1, the T_10_ for PCL-PU is 309 °C, which reduced to 276 °C after the addition of 5–10% chitosan, and again increased to 318 °C. This change could be due to the increased interaction of chitosan and PCL-diol, which stabilizes the polymer system by means of increased cross-linking points. A reduction to 255 °C was found in a case of 20% chitosan content, suggesting phase separation. The residual content after thermal degradation, which is a nondegradable carbon residue of chitosan, was also found to increase with an increase in the content of chitosan. Furthermore, it can be derived from the TGA analysis that the presence of chitosan, which has a lower degradation temperature, had no adverse effect on the thermal stability of the polyurethane biocomposite. The crosslinking between the PCL diol and chitosan molecule could be responsible for the stabilization of the polymer system. The crosslinking between the PCL diol and chitosan was also confirmed by the gel content analysis.

### 3.7. Gel Content of Polyurethane Biocomposites

The samples were suspended in chloroform in order to dissolve the non-crosslinked content, and the insoluble content was measured. A gel content of ~25.4%, which increased to 81.2% with an increase in the content of chitosan from 0% to 20% (Figure 8), was found for the PCL-PU. This improvement in the gel content suggests an enhancement in cross-linking that reduced the solubility of the polymer system. The presence of chitosan provided the cross-linking points which led to an increase in the gel content.

### 3.8. Shape Memory Effect of PCL Based Polyurethane Biocomposites

Figure 9 demonstrates the shape memory effect of the prepared polyurethane biocomposite in a dry condition where the polyurethanes were initially fixed by heating them to 50 °C followed by cooling them to room temperature. It was found that the time required for shape recovery after the fixing was quite similar for the PCL-PU and PCL-PU-Chitosan biocomposites. The shape recovery of the biocomposite was also demonstrated in a wet condition, which is shown in Figure 10. A random shaped polymer sheet was punched with circular shapes and mechanically stretched at 50 °C followed by cooling to room temperature. The mechanical stretching led to a permanent deformation of the circled shapes, which were recovered by approximately 100% by placing the sample in warm water at 50 °C. For the quantitative analysis of the shape memory behavior of the polyurethane biocomposite, the shape fixing ratio and shape recovery ratio was measured and is presented in Table 2. It is seen that the shape fixing ratio improved from ~53% to approximately 100% after the incorporation of 10% chitosan. Similarly, the shape recovery ratio was found to be improved from ~88% to ~100%.

The mechanical analysis revealed that the toughness of the materials was drastically improved, which could be the result of the increased crosslinked nodes in the polymer system. The shape recovery of the prepared material from an elongated (~500% strain) or permanently deformed condition is shown in Figure 11.

The coiling and uncoiling of the PCL chains could be responsible for the shape memory ability, which greatly corresponds to the crosslink content in the polymer matrix. It was found that the polyurethane biocomposite with chitosan retains its temporary shape even after the removal of stress and sustained restraint. The developed restraint in the shape memory polyurethane biocomposite due to tensile pull is possibly released during thermal softening around the melting of the PCL crystallites. An upsurge in the temperature triggers the melting of PCL crystallites which are formed due to a stress induced crystallization phenomenon during tensile pull, and the material tends to recover its original shape. The chitosan nanoflakes work as rigid crosslinking nodes covalently bonded between extendable PCL chains. These cross-linking nodes could be responsible for putting back polymer chains in response to an external stimulus. On the other hand, the formation of crystallites facilitates a proficient and quick formation of the switching domain, as observed in the DSC analysis (the melting of PCL crystallites in the range of 40–50 °C) and also confirmed from the X-ray analysis.

XRD analysis of the stretched and non-stretched (PCL-PU-chitosan) samples is shown in Figure 3b. It can be seen that the intensity of the characteristic peaks for PCL at 21.3 ° and 23.6 ° is enhanced, suggesting that the crystallinity of the biocomposite is increased. The enhancement in the crystallinity could be the consequence of stress induced crystallization during tensile work. The uncoiling of coiled PCL chains, and arrangement of the same, increases the crystallinity of the biocomposite and possibly enhances the shape memory even after permanent deformation. Also, the shape recovery phenomenon of PCL based chitosan polyurethane biocomposite is demonstrated in Appendix A.

## 4. Conclusions

We have successfully demonstrated the fabrication of a PCL based shape memory polyurethane biocomposite via in situ incorporation of chitosan. The developed biocomposite with 10% chitosan content had ~130% enhancement in ultimate tensile strength and more than 500% elongation at break in comparison to pristine PCL-PU. The electron microscopic analysis confirmed the homogeneous distribution of chitosan flakes with ~150 nm width in the polymer matrix. X-ray and DSC analysis confirmed the presence of PCL crystallites, which may act as a switching point for the shape memory behavior. X-ray diffraction analysis also confirmed the development of crystallites, resulting from a stress induced crystallization process during tensile pull, which may retain the shape and melting of the crystallites, thereby stimulating shape recovery (with ~100% shape recovery ratio), even after permanent deformation. The biodegradable polyurethane biocomposite also demonstrates relatively higher thermal stability (T_max_ at ~360 °C). Overall, the fabricated shape memory polyurethane biocomposite possesses a unique shape memory behavior, even after permanent deformation to more than a 500% strain, which has extreme potential in several biomedical applications.

## Figures and Tables

**Figure 1 nanomaterials-09-00225-f001:**
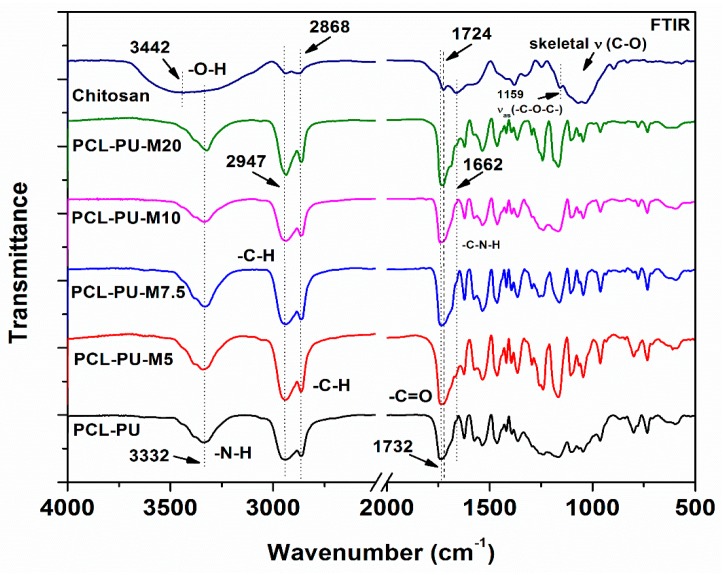
FTIR spectra of Chitosan, polycaprolactone-polyurethane (PCL-PU), and PCL-PU-Chitosan biocomposites.

**Figure 2 nanomaterials-09-00225-f002:**
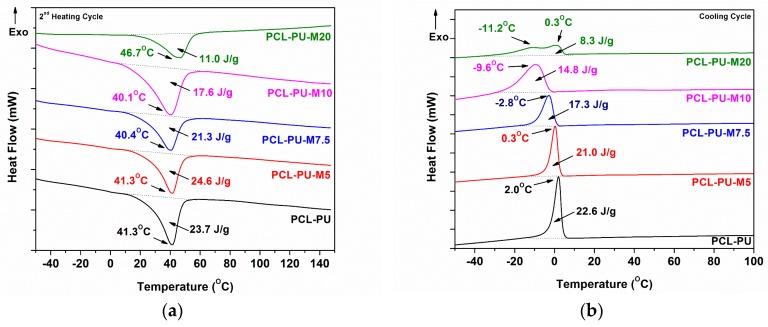
Differential scanning calorimetry (DSC) thermograms: (**a**) Second heating and (**b**) cooling cycle of the PCL-PU and PCL-PU-Chitosan biocomposites.

**Figure 3 nanomaterials-09-00225-f003:**
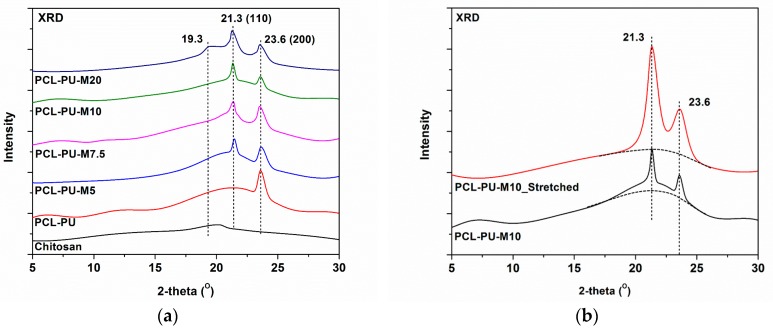
(**a**) X-ray diffraction of the PCL-PU and PCL-PU-Chitosan biocomposite and (**b**) X-ray diffraction comparison of the stretched and non-stretched biocomposite sample.

**Figure 4 nanomaterials-09-00225-f004:**
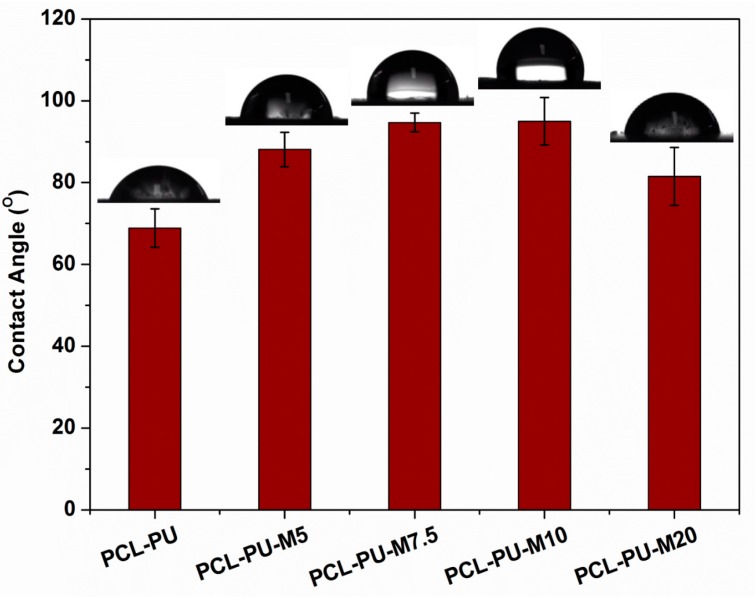
Contact angle measurement of PCL-PU and PCL-PU-Chitosan biocomposites.

**Figure 5 nanomaterials-09-00225-f005:**
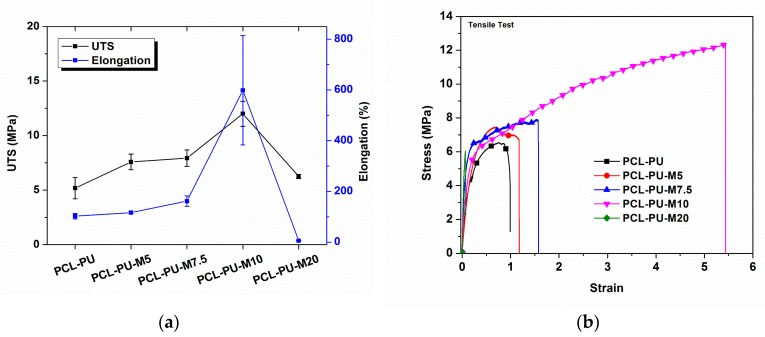
(**a**) Ultimate tensile strength (UTS) and elongation at break of the PCL-PU and PCL-PU-Chitosan polyurethane biocomposites and (**b**) representative stress vs. strain curve for the same.

**Figure 6 nanomaterials-09-00225-f006:**
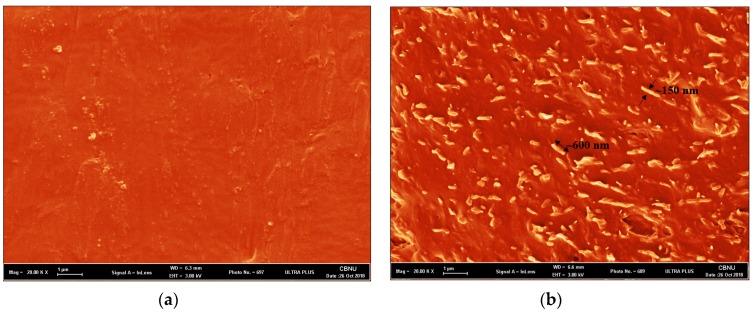
Representative field emission scanning electron microscope (FESEM) micrographs of (**a**) the PCL-PU and (**b**) the PCL-PU-Chitosan biocomposite (Scale bar: 1 µm).

**Figure 7 nanomaterials-09-00225-f007:**
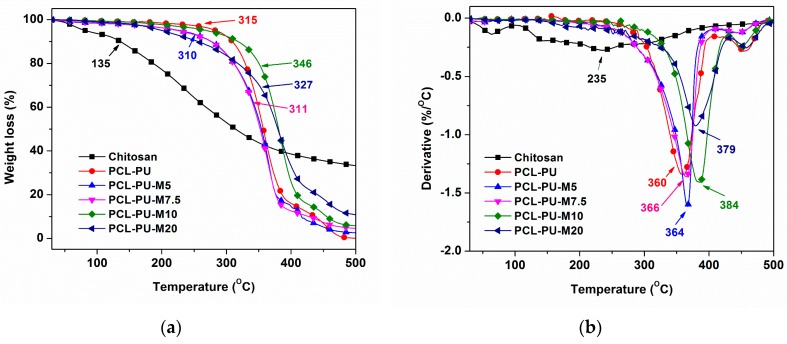
(**a**) Weight loss and (**b**) derivative weight loss of PCL-PU and PCL-PU-Chitosan biocomposite against temperature.

**Figure 8 nanomaterials-09-00225-f008:**
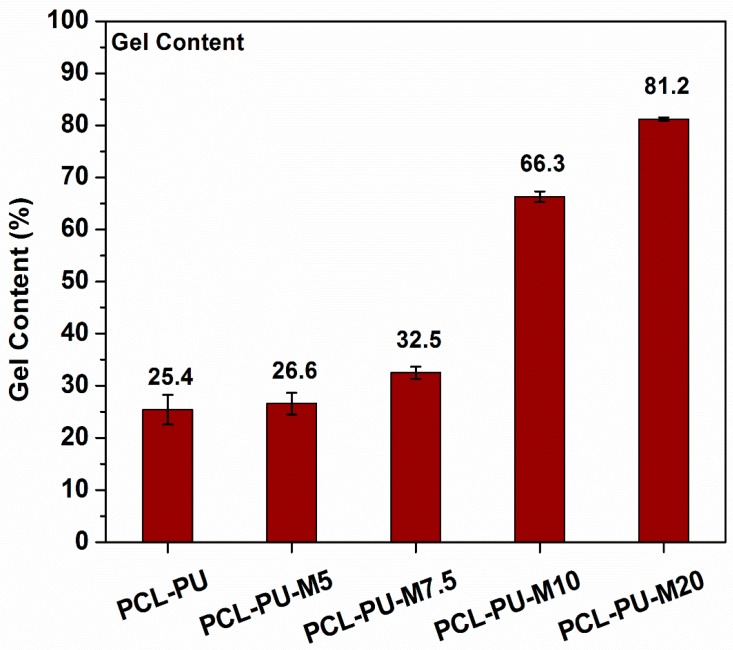
Gel content in the PCL-PU and PCL-PU-Chitosan biocomposites in chloroform.

**Figure 9 nanomaterials-09-00225-f009:**
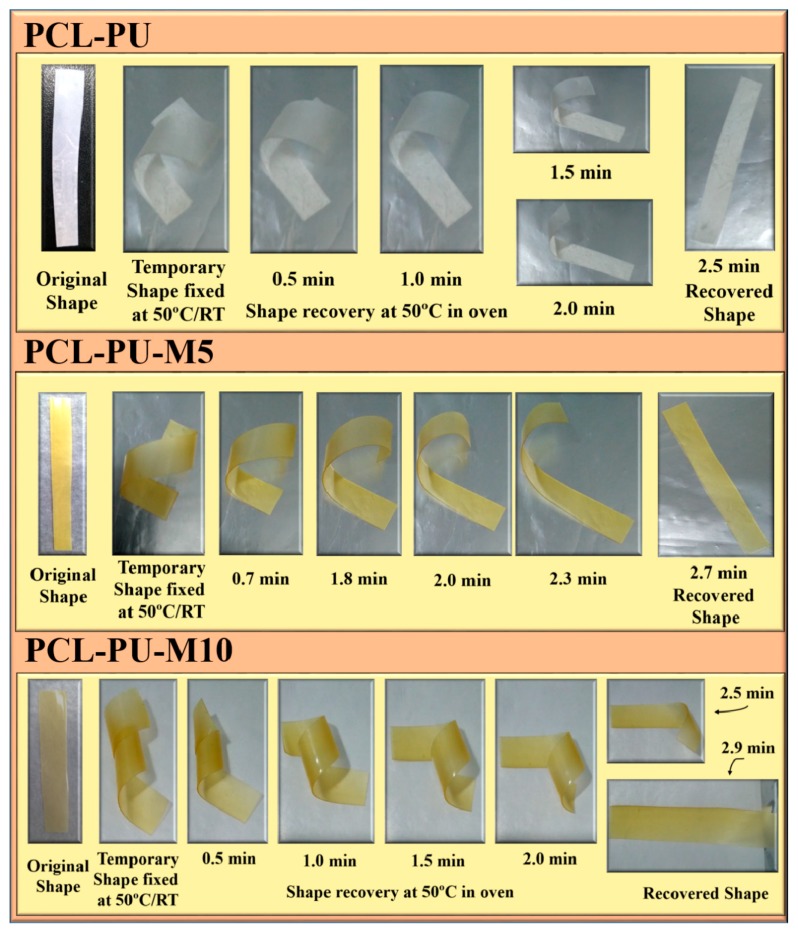
Demonstration of the shape memory effect in the PCL-PU-Chitosan biocomposite in a dry condition.

**Figure 10 nanomaterials-09-00225-f010:**
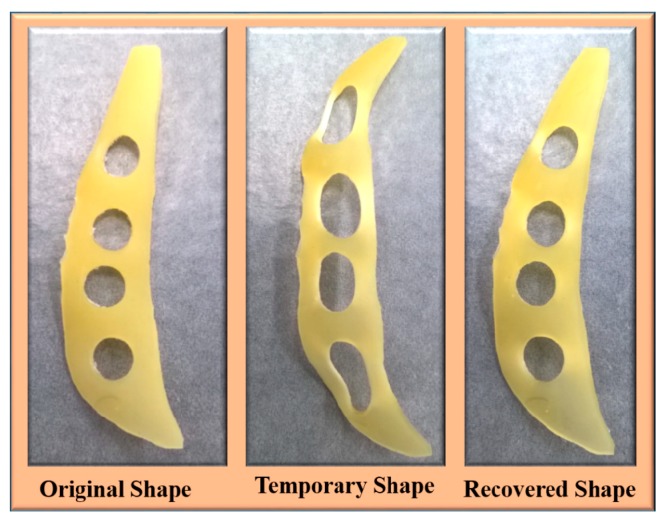
The shape memory ability of the prepared random shaped chitosan polyurethane composite for grip application (Temporary shape at room temperature using tensile force; recovered shape in water (wet condition) at 50 °C within 5 s).

**Figure 11 nanomaterials-09-00225-f011:**
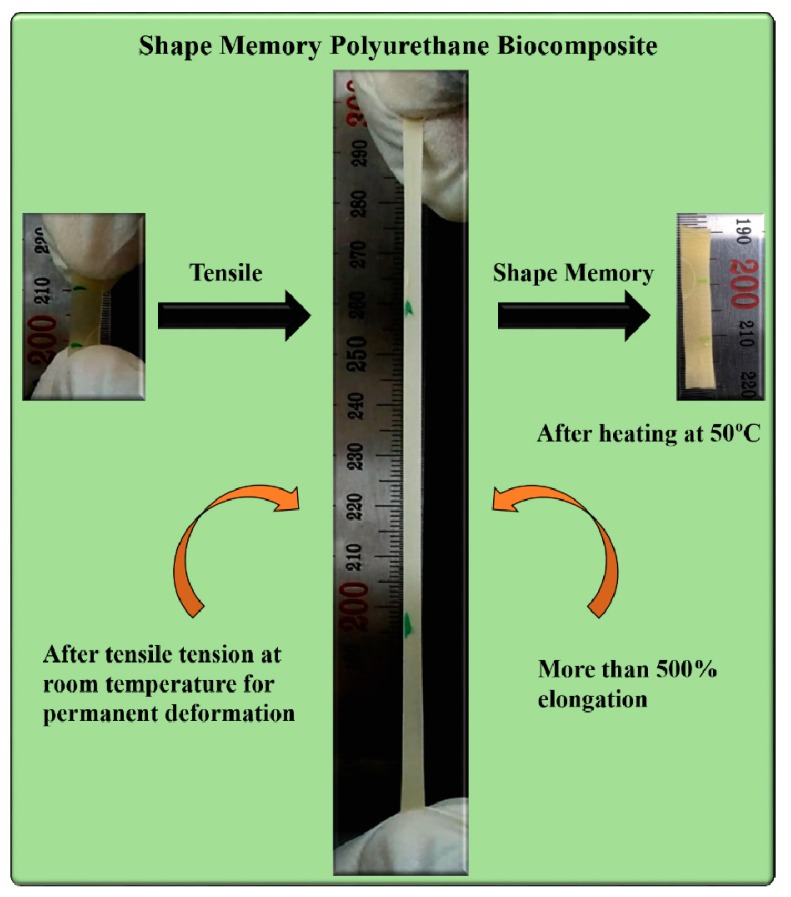
Shape memory behavior of the polyurethane chitosan biocomposite after tensile tension.

**Table 1 nanomaterials-09-00225-t001:** Thermal transitions and percentage residue for PCL-PU and PCL-PU-Chitosan biocomposites.

Sample Name	T_onset_ (°C)	T_max_ (°C)	T_10_ (°C)	T_50_ (°C)	Residue (%)
**Chitosan**	135	235	135	314	33.2
**PCL-PU**	315	360	309	357	0.0
**PCL-PU-M5**	310	364	276	354	2.6
**PCL-PU-M7.5**	311	366	276	352	4.7
**PCL-PU-M10**	346	384	318	381	5.8
**PCL-PU-M20**	327	379	255	379	10.8

**Table 2 nanomaterials-09-00225-t002:** Shape fixing and shape recovery ratios of the PCL-PU and PCL-PU-Chitosan polyurethane biocomposites.

Polyurethane Samples	Shape Fixing Ratio (%)	Shape Recovery Ratio (%)
**PCL-PU**	53.1 ± 6.2	88.6 ± 1.3
**PCL-PU-M5**	81.2 ± 2.6	90.8 ± 1.2
**PCL-PU-M7.5**	93.4 ± 5.0	98.5 ± 0.5
**PCL-PU-M10**	99.5 ± 0.4	99.3 ± 0.7
**PCL-PU-M20**	Not stretchable	Not stretchable

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
