# Peer review of "Shape Memory Polyurethane Biocomposites Based on Toughened Polycaprolactone Promoted by Nano-Chitosan"

_nanomaterials, 2019, doi:10.3390/nano9020225_

Round 1

Reviewer 1 Report

The manuscript entitled “Shape memory polyurethane biocomposites based on toughened polycaprolactone promoted by nano-chitosan” describes the preparation of shape memory polyurethanes reinforced with chitosan flakes.

This work is well-structured, which facilitates its reading and interpretation. In addition, the paper presents very interesting work and allows to advance a little more in its research field.

The present manuscript should be published after the authors take into account the following remarks

A)    The authors indicate that the increase in mechanical properties for load polymers could be due to the high aspect ratio of the flakes. However, the main reason is the increase in cross-linking points produced with chitosan flakes. In fact, the authors have observed that as the content of chitosan increases, the gel content also increases. Please correct.

B)    The authors calculated that the gel content of PCL-PU was 25.4%. However, in the experimental part, the authors indicate that the NCO / OH ratio is 1: 1, that is, the polymer should be completely soluble. Could the authors explain this fact?

Author Response

Reviewer’s Comments

Author’s Response

Reviewer   1: The manuscript entitled “Shape memory   polyurethane biocomposites based on toughened polycaprolactone promoted by   nano-chitosan” describes the preparation of shape memory polyurethanes   reinforced with chitosan flakes.

This work is well-structured, which   facilitates its reading and interpretation. In addition, the paper presents   very interesting work and allows to advance a little more in its research   field.

The present manuscript should be   published after the authors take into account the following remarks

01.   

The authors indicate that the increase   in mechanical properties for load polymers could be due to the high aspect   ratio of the flakes. However, the main reason is the increase in   cross-linking points produced with chitosan flakes. In fact, the authors have   observed that as the content of chitosan increases, the gel content also   increases. Please correct.

The sentence related to the aspect ratio   is omitted from the manuscript as per suggestion from reviewer.

02.   

The authors calculated that the gel content   of PCL-PU was 25.4%. However, in the experimental part, the authors indicate   that the NCO / OH ratio is 1: 1, that is, the polymer should be completely   soluble. Could the authors explain this fact?

It was written mistakenly as 1:1 instead   of 1:1.1. Correction is incorporated in the experimental section.

Reviewer 2 Report

The article is well written and supported with tests and data adequate for the scope of this paper.

Anyway, the reviewer is not comfortable with the abuse of use of “biocompound” especially when referred to a polyurethane obtained from oil-based material. The use in biomedical application is postulated, but not demonstrated. Equally, the biodegradation is supposed but not demonstrated, similarly the non-toxicity is not verified. Being outside of the scope of the article, we can waive on the legitimate presence of these data, but the reviewer invites the authors to do not abuse of this kind of assertions without sufficient basis.

Apart, small glitches in the manuscript have to be corrected. i.e.:

Line 41: “diisocynate” instead of “diisocyanate”

Line 75 “near thermal degradable temperature” instead of “near degradation temperature”

Line 85 “bone” instead of “bones”

Line 126 “two necked round bottomed flask” instead of “two-neck round-bottom flask”

Line 154 the dot of 1.540593A is not in the correct format

Line 258 there is a mismatch of font in “ultimate tensile strength”

Line 358 “retain” instead of “retains”

Line 431, 437,451470, 480, 489 pay attention to the upper-case letter of the abbreviations.

Author Response

Reviewer   2: The article is well written and   supported with tests and data adequate for the scope of this paper.

Anyway, the reviewer is not comfortable   with the abuse of use of “biocompound” especially when referred to a   polyurethane obtained from oil-based material. The use in biomedical   application is postulated, but not demonstrated. Equally, the biodegradation   is supposed but not demonstrated, similarly the non-toxicity is not verified.   Being outside of the scope of the article, we can waive on the legitimate   presence of these data, but the reviewer invites the authors to do not abuse   of this kind of assertions without sufficient basis.

Apart, small glitches in the manuscript   have to be corrected. i.e.:

Line 41: “diisocynate” instead of   “diisocyanate”

Sentence is corrected as per suggestion.  

Line 75 “near thermal degradable   temperature” instead of “near degradation temperature”

Correction in the sentence is done as   per reviewer suggestion.

Line 85 “bone” instead of “bones”

Word is corrected as per suggestion by reviewer.

Line 126 “two necked round bottomed   flask” instead of “two-neck round-bottom flask”

Line is corrected as per suggestion.

Line 154 the dot of 1.540593A is not in   the correct format

Format is corrected as per suggestion.

Line 258 there is a mismatch of font in   “ultimate tensile strength”

Font size and formatting is corrected as   per suggestion.

Line 358 “retain” instead of “retains”

Spelling is corrected as per suggestion.

Line 431, 437,451, 470, 480, 489 pay   attention to the upper-case letter of the abbreviations.

Abbreviations in references are   corrected as per suggestion.

Reviewer 3 Report

NANOMATERIALS-438646

Shape Memory Polyurethane Biocomposites Based on Toughened Polycaprolactone Promoted by Nano-Chitosan

This manuscript was described about fabrication of thermos-responsive shape-memory biodegradable polycaprolactone (PCL) based polyurethane biocomposite via in situ incorporation of chitosan. The incorporation of chitosan enhanced the mechanical properties and the composite showed good shape memory behaviors. The manuscript is recommended minor revision. Other comments are as follows;

1. Line 136; “All samples was prepared by maintaining the NCO:OH final ratio as 1:1.”

Does numbers of OH group include OH group of chitosan?

2. Line 219; “The presence of chitosan possibly hinders the PCL chain mobility restricting the orientation of chains”

How about the glass transition temperature of PCL based polyurethane and the biocomposite?

3. Line 219; “The melting temperature of biocomposite is higher  in case of 20% chitosan content which could be attributed to the phase separation of PCL and chitosan molecules”

Because of SEM observation, it is clear that PCL and chitosan are heterogeneous in all composites. Was chitosan aggregated in case of 20% chitosan content?

4. Line 245; “As the chitosan flake content is increasing in the biocomposite, the contact angle is gradually increasing possibly due to the reduction in surface free energy.”

Why do surface free energy reduce by the incorporation of chitosan?

5. Figure 6; Please provide scale bar.

6. Line 356; “The coiling and uncoiling of the PCL chains can be responsible for the shape memory ability which highly corresponds to the crosslink content into the polymer matrix.”

In the biocomposite, OH group of chitosan reacted with 6-hexamethylenediisocynate to form crosslinking network. On the other hand, PCL-based polyurethane was prepared without chitosan to form linear chain. Why the PCL-based polyurethane show shape-memory behaviors?

Author Response

Reviewer   3: This manuscript was described about   fabrication of thermo-responsive shape-memory biodegradable polycaprolactone   (PCL) based polyurethane biocomposite via in situ incorporation of chitosan.   The incorporation of chitosan enhanced the mechanical properties and the   composite showed good shape memory behaviors. The manuscript is recommended   minor revision. Other comments are as follows;

01.   

Line 136; “All samples was prepared by   maintaining the NCO:OH final ratio as 1:1.” Does numbers of OH group include   OH group of chitosan?

Yes, the number of OH group include   functional group of chitosan. It was written mistakenly as 1:1 instead of 1:1.1.   Correction is incorporated in the experimental section.

02.   

Line 219; “The presence of chitosan   possibly hinders the PCL chain mobility restricting the orientation of   chains”. How about the glass transition temperature of PCL based polyurethane   and the biocomposite?

We have done the differential scanning calorimetric   analysis of the samples and we found that the deflection of the curve for   glass transition temperature was significantly invisible due to which it was   not possible to measure therefore, we have not reported it.

03.   

Line 219; “The melting temperature of biocomposite   is higher  in case of 20% chitosan   content which could be attributed to the phase separation of PCL and chitosan   molecules”. Because of SEM observation, it is clear that PCL and chitosan are   heterogeneous in all composites. Was chitosan aggregated in case of 20%   chitosan content?

We know from the several studies [1,   2]   done by scientific community that the compatibility and miscibility of the   chitosan with hydrophobic polymer is poor in higher amount which may lead to   aggregation of the chitosan in polymer matrix resulted in phase separation.

04.   

Line 245; “As the chitosan flake content   is increasing in the biocomposite, the contact angle is gradually increasing   possibly due to the reduction in surface free energy.” Why do surface free   energy reduce by the incorporation of chitosan?

It is known that there is no direct   method to measure the surface free energy however, contact angle might be   used for the measurement of the same. In the current work, we have found   increment in the contact angle after incorporation of chitosan which may   resulted due to the increased surface roughness and reduction in the surface   free energy. Reduction in the surface free energy might be the result of   increased cross linked content.

05.   

Figure 6; Please provide scale bar.

Incorporated in the manuscript as per   suggestion from the reviewer.

06.   

Line 356; “The coiling and uncoiling of   the PCL chains can be responsible for the shape memory ability which highly   corresponds to the crosslink content into the polymer matrix.” In the   biocomposite, OH group of chitosan reacted with 6-hexamethylenediisocynate to   form crosslinking network. On the other hand, PCL-based polyurethane was   prepared without chitosan to form linear chain. Why the PCL-based   polyurethane show shape-memory behaviors?

It is known that the shape memory   behavior is mainly associated with the melting of crystals (in the case of   polycaprolactone polyurethane) formed during deformation and fixation of the   samples. However, in the case of chitosan incorporated polyurethane, the   enhancement in the behavior was found which may be the result of crosslink   content.

References

1.        Ye, J.-R., et al., Turning the chitosan surface from hydrophilic to hydrophobic by layer-by-layer electro-assembly. RSC Advances, 2014. 4(102): p. 58200-58203.

2.        Guan, X., et al., Chitosan-graft-poly(ϵ-caprolactone)s: An optimized chemical approach leading to a controllable structure and enhanced properties. Journal of Polymer Science Part A: Polymer Chemistry, 2007. 45(12): p. 2556-2568.